# Freeway Traffic Congestion Reduction and Environment Regulation via Model Predictive Control

**Juan Chen [1,2,*]** **, Yuxuan Yu [1] and Qi Guo [1]**

1    SHU-UTS SILC Business School, Shanghai University, Shanghai 201899, China; gy6667777@163.com (Y.Y.); iguoqi@i.shu.edu.cn (Q.G.)
2    Smart City Research Institute, Shanghai University, Shanghai 201899, China
*    Correspondence: chenjuan82@shu.edu.cn; Tel.: +86-10-69980028-55051

**Abstract:** This paper proposes a model predictive control method based on dynamic multi-objective optimization algorithms (MPC_CPDMO-NSGA-II) for reducing freeway congestion and relieving environment impact simultaneously. A new dynamic multi-objective optimization algorithm based on clustering and prediction with NSGA-II (CPDMO-NSGA-II) is proposed. The proposed CPDMO-NSGA-II algorithm is used to realize on-line optimization at each control step in model predictive control. The performance indicators considered in model predictive control consists of total time spent, total travel distance, total emissions and total fuel consumption. Then TOPSIS method is adopted to select an optimal solution from Pareto front obtained from MPC_CPDMO-NSGA-II algorithm and is applied to the VISSIM environment. The control strategies are variable speed limit (VSL) and ramp metering (RM). In order to verify the performance of the proposed algorithm, the proposed algorithm is tested under the simulation environment originated from a real freeway network in Shanghai with one on-ramp. The result is compared with fixed speed limit strategy and single optimization method respectively. Simulation results show that it can effectively alleviate traffic congestion, reduce emissions and fuel consumption, as compared with fixed speed limit strategy and classical model predictive control method based on single optimization method.

**Keywords:** freeway transportation; congestion control; environment impact; dynamic multi-objective optimization; model predict control; clustering and prediction

## 1. Introduction

As people's demand for driving increases, freeways have rapidly reached saturation and traffic congestions occur frequently. In China, reoccurred congestion takes up a large proportion, contributing to wasting of people's time, as well as economic loss. In addition, due to serious environmental pollution and resources shortages, it is important to pay attention to emissions and fuel wastage resulting from traffic congestion. Therefore, it is important to handle congestion through reasonable freeway control methods, considering the limited resources. This paper proposes a more scientific and effective control method to alleviate congestion, reduce pollution and energy wastage.

This paper mainly focuses on the freeway control in a real network with an on-ramp. It uses METANET model to simulate traffic behaviors of the freeway, and VT-micro for estimating emissions and fuel consumption. The authors of [1] provide a general framework to integrate these two kinds of models, the macroscopic traffic flow model METANET and the microscopic emission and fuel consumption model VT-micro, resulting in the so called the "VT-macro" model. Most of the papers consider the variable speed limit (VSL) as continuous values, however, in this paper, the VSL values are treated as discrete values following the ideas in [2], which, not only shows a good performance, but also keeps the computation time reduced.

In this article, a model predictive control method based on dynamic multi-objective optimization algorithm, named as MPC_CPDMO-NSGA-II, is designed and tested under the simulation environment. Furthermore, in order to provide more reasonable and effective freeway control solutions, a new dynamic multi-objective optimization algorithm based on clustering and prediction strategy, named as CPDMO-NSGA-II, is proposed to realize on-line optimization in model predictive control at each control step.

The aim of the paper is to relieve freeway congestion, reduce fuel consumption and emission simultaneously based on a model predictive control method. Considering that the essence of the freeway control problem mentioned above is a multi-objective control problem, four performance indicators are considered in this paper, which is to minimize the total time spent, total distance, fuel consumption and emissions simultaneously. The classical way of dealing with multiple conflict objectives in the model predictive control method is to sum up the different objectives into a single comprehensive performance indication by a predefined threshold. Its aim is to translate the multi-objective problem into a single objective control problem. Although the weight sum method is easy to realize, the threshold must be predefined according to expert experience and better trade-off among different conflict objectives cannot be guaranteed, especially when the search space is non-linear, non-convex, discontinuous, etc. Therefore, it is meaningful to use a multi-objective optimization algorithm to optimize the multiple conflict objectives simultaneously in the model predictive control method. At the same time, since the model predictive control method has the characteristics of model prediction, receding horizon and feedback correction, it is not enough to replace the weight sum method with the multi-objective optimization algorithm and solve it using the classical static multi-objective optimization algorithm. Therefore, a new dynamic multi-objective genetic algorithm, CPDMO-NSGA-II, is proposed in this paper to realize on-line multi-objective optimization in the model predictive control. There are two purposes for replacing the weight sum method with the dynamic multi-objective optimization algorithm, first is to enable the algorithm to adapt to the environmental variation and converge to the optimal solution at each control step as soon as possible and second is to realize better trade-off among different multiple objectives.

The contributions of this paper are as follows: (1) Both freeway congestion and environment impact are optimized simultaneously based on model predictive control method. (2) A dynamic multi-objective genetic algorithm is proposed to realize better trade-off among multiple objectives in the model predictive control method. (3) The proposed model predictive control method is tested under the simulation environment and compared with fixed time speed strategy and single objective optimization algorithm.

The remainder of this paper is organized as follows. Section 2 provides the literature review. The macroscopic traffic flow model METANET, the VT-micro model and the integrated VT-macro model are described in Section 3. Section 4 discuss the proposed model predictive control method based on dynamic multi-objective optimization algorithm, MPC_CPDMO-NSGA-II. Section 5 describes the problem formulation in detail. Section 6 discuss the simulation result of the freeway control problem. Section 7 provides conclusions and future works.

## 2. Literature Review

Model predictive control (MPC) is a new control algorithm that was proposed in the late 1970s. It is a control algorithm based on model prediction [3]. When a traffic light is placed at the on-ramp of a freeway, a ramp metering is set-up. The traffic light turns over between the red and the green phase. The number of vehicles that enters the freeway through the on-ramp is controlled by the variation of the timing of the red and the green phases. In order to keep the traffic density below the critical density, the inflow of vehicles onto the freeway is limited by the ramp metering set-up. Then, the traffic breakdown and congestion is avoided. A waiting queue of vehicles is formed at the on-ramp whenever the traffic demand is larger than the number of cars allowed to enter the freeway [4].

MPC is a commonly-used method in freeway control [5,6]. It is usually solved through resilient back propagation [7] and sequence quadratic programming [8]. MPC is quite often designed for linear or linearized system with linear constraints in order to solve optimization problem easily. However, the on-line optimization problem in MPC with the characteristics of non-linearity, non-convex and complicated constraints is considered in some existed research recently. For example, the authors of [2,9], introduce genetic algorithm (GA) into MPC to realize on-line optimization. In order to acquire better performance for freeway control, some studies use multiple indicators for evaluation. These performance indicators are considered as objectives in optimization. Some of them are conflicts with each other [10,11]. However, most of the papers adopt weighted sum methods to transfer the multi-objective problem into single-objective problem [12,13]. However, it performs worse in solving multi-objective problems. Moreover, repeated experiments are necessary to determine proper weighs. Thus, the authors of [14] adopt a fast and elitist multi-objective genetic algorithm (NSGA-II) [15], which is verified as an effective algorithm to alleviate congestions. Considering that the essence of the on-line optimization problem in MPC at each control is a dynamic multi-objective problem, a new solution needs to be provided by the controller in each control step. Therefore, static multi-objective optimization algorithms, such as NSGA-II, cannot rapidly respond to the change of environment. Thus, dynamic multi-objective optimization algorithm is used to realize on-line optimization in model predictive control at each control step.

As for dynamic multi-objective optimization algorithm (DMOAs), it is important to maintain or increase population diversity to search for new Pareto front in the current environment. It spends a long time to be converged. Prediction is an effective strategy to shorten the time of convergence [16]. It tracks the changed Pareto front through historical individuals to provide guidance direction in evolution, which also increases the diversity of population. The authors of [17] first employ prediction models in economics, such as AR and VAR. But the premise is that historical data have statistical characteristics. The authors of [18] then propose four general methods, RND, VAR, PRE and V&P. They are easier to operate than AR and VAR. However, the performance of distribution they considered and the historical information they offered are limited. The authors of [19,20] design a strategy called clustering prediction. The individual elements of prediction are replaced with centers of clustering, which are regarded as the representatives of the Pareto front. But the performance is related to the method of clustering.

We will discuss model predictive control method based on dynamic multi-objective optimization algorithm in the next section. Traditional, the minimum reference deviation, minimum variation or cost of some parameters, minimum control action is used as the multiple objectives in model predictive control, and the multiple objectives are combined as a single objective problem by the weight sum method. In order to get optimal solutions, the weight used in the objectives in model predictive control is decided by expert's experience or trial-and-error normally, while in some cases, the weight is adjusted dynamically. This study makes the first attempt to combine model predictive control with a dynamic multi-objective genetic algorithm to find Pareto optimal solutions for freeway congestion control problem. Then TOPSIS method is adopted to select the solution to be implemented. In order to assess the performance of the proposed approach [21], the proposed algorithm is tested under the simulation environment originated from a real freeway network in Shanghai with one on-ramp. Compared with classical MPC approaches that deal with multiple objectives by the weighted sum approach, the model predictive control method based on dynamic multi-objective genetic algorithm can better fulfill goals of alleviating traffic congestion, reducing emissions and fuel consumption although it may also be computationally expensive. With the rapid development of multi-objective evolutionary algorithms, our study suggests the potential of model predictive control method based on dynamic multi-objective genetic algorithm to deal with a wide range of freeway congestion control problems in the future.

In this paper, the METANET model is used to predict the traffic states, the integrated VT-macro model is used to calculated total emissions and fuel consumptions. Therefore, we will discuss the METANET model and VT- macro model first, then the multi-objective control problem solved by

MPC_CPDMO-NSGA-II algorithm and related constraints will be described later. Finally, the simulation result will be discussed. The acronym List are given in Appendix A.

## 3. Traffic Flow Models

### 3.1. METANET Model

METANET is a widely used macroscopic second-order traffic flow model. It can simulate traffic behaviors at specific time and location of the highway with arbitrary topology and characteristics, including fundamental segments, on-ramp, and intersections [2]. Besides, it can replicate traffic waves at bottlenecks, as well as capacity degradation caused by congestions. It discretizes the traffic flow temporally and spatially. The time interval is $T_s$. As shown in Figure 1, the highway $m$ is spatially divided into $N_m$ segments with equal length $L_m$. Average density $\rho_{m,i}(k)$, speed $v_{m,i}(k)$ and flow $q_{m,i}(k)$ are three basic traffic variables. $k = 0, 1, \ldots, K$, where $K$ is the timeline. METANET with VSL and RM can be expressed as follows:

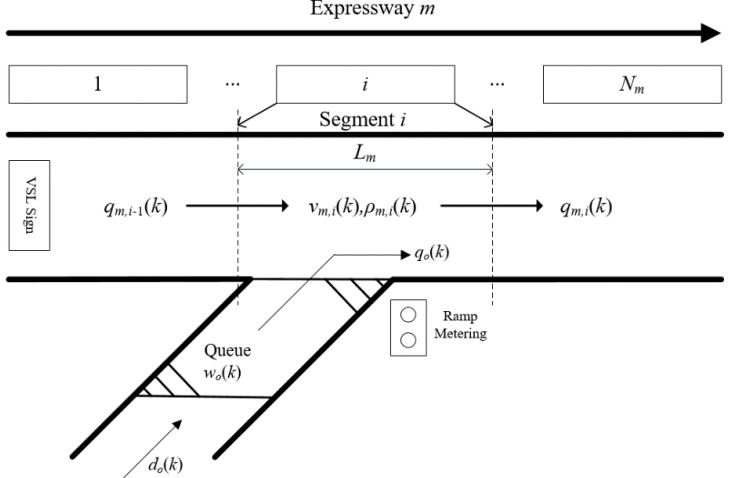

**Figure 1.** Discretized highway link.

The average density is calculated as follows:

$$\rho_{m,i}(k+1) = \rho_{m,i}(k) + \frac{T_s}{\lambda_m L_m}[q_{m,i-1}(k) - q_{m,i}(k) + q_o(k)], \tag{1}$$

where $\lambda_m$ is the number of lanes; $\rho_{m,i}(k)$ is the density of segment $i$ at $kT_s$; and $q_o(k)$ is the flow entered into mainline from on-ramps.

The average speed is calculated as follows:

$$\begin{aligned}
v_{m,i}(k+1) &= v_{m,i}(k) + \frac{T_s}{\tau}\{V[\rho_{m,i}(k)] - v_{m,i}(k)\} + \frac{T_s}{L_m}v_{m,i}(k)[v_{m,i-1}(k) \\
&- v_{m,i}(k)] - \frac{vT_s}{\tau L_m}\frac{\rho_{m,i}(k+1) - \rho_{m,i}(k)}{\rho_{m,i}(k) + \kappa} - \frac{\delta T_s v_{m,i}(k)q_o(k)}{L_m \lambda_m[\rho_{m,i}(k) + \kappa]}
\end{aligned} \tag{2}$$

where $v$, $\tau$, $\delta$ and $\kappa$ are parameters of the model. $v_{m,i}(k)$ represents the speed of segment $i$ at $kT_s$. $V[\rho_{m,i}(k)]$ is average desired speed. $-\frac{\delta T_s v_{m,i}(k)q_o(k)}{L_m \lambda_m[\rho_{m,i}(k) + \kappa]}$ denotes the decreased of speed resulted by entered vehicles from on-ramps.

The average flow is calculated as follows:

$$q_{m,i}(k) = \lambda_m \rho_{m,i}(k) v_{m,i}(k). \tag{3}$$

The average desired speed is calculated as follows:

$$V[\rho_{m,i}(k)] = \min(v_{f,m}\exp[-\frac{1}{\alpha_m}(\frac{\rho_{m,i}(k)}{\rho_{cr,m}})^{\alpha_m})], (1+a)v_{ctrl,i}), \tag{4}$$

where $\alpha_m$ is parameters of METANET; $a$ is non-compliance rate; $\rho_{cr,m}$ is critical density; $v_{f,m}$ is free speed; and $v_{ctrl,i}$ is the value of VSL executed on segment $i$.

Due to the effects of congestion, capacity of origins or traffic lights, if demands exceed the capacity of some segment received, queue will be formed, marked as $w_o$. Take the on-ramp $o$ located in segment $i$ for example, the queue length of origins is calculated as follows:

$$w_o(k+1) = w_o(k) + T_s \cdot [d_o(k) - q_o(k)] \tag{5}$$

$$q_o(k) = r_o(k)\hat{q}_o(k) \tag{6}$$

$$\hat{q}_o(k) = \min\{\hat{q}_{o,1}(k), \hat{q}_{o,2}(k)\} \tag{7}$$

$$\hat{q}_{o,1}(k) = d_o(k) + w_o(k)/T_s \tag{8}$$

$$\hat{q}_{o,2}(k) = Q_o\min\left\{1, \frac{\rho_{\max} - \rho_{m,i}(k)}{\rho_{\max} - \rho_{cr,m}}\right\}, \tag{9}$$

where $\rho_{\max}$ is the maximal density. The outflow $q_o(k)$ depends on the enforced RM and traffic variables of segments in which on-ramps located. If $r_o(k) \in [r_{\min}, 1]$, it denotes the RM rate. If there is RM, the flow $r_o(k)$ ultimately left on-ramp $q_o(k)$ in $[kT_s, (k+1)T_s]$ is defined as a portion of maximum outflow $\hat{q}_o(k)$ without RM. $r_o(k) = 1$ denotes RM is absent, otherwise $r_o(k) < 1$. If $\hat{q}_{o,1}(k) < \hat{q}_{o,2}(k)$, $\hat{q}_o(k)$ is determined by the demand $d_o$ at $kT_s$. Otherwise, it depends on capacity $Q_o$ when the density of the mainline is under-critical, i.e., $\rho_{m,1}(k) < \rho_{cr,m}$, or the reduced capacity result from congestion of the mainline, i.e., $\rho_{m,1}(k) > \rho_{cr,m}$.

## 3.2. Integrating METANET Model with VT-micro Model

In this paper, the VT-macro model proposed in [1], which integrated the METANET model and the VT-micro, is used to calculated total emissions and fuel consumptions.

In order to balance the performances of computation time and accuracy, the VT-micro model proposed by the authors of [22] is used to estimate emissions and fuel consumption in this paper. Figure 2 describes the temporal and spatial traffic flow in the METANET model.

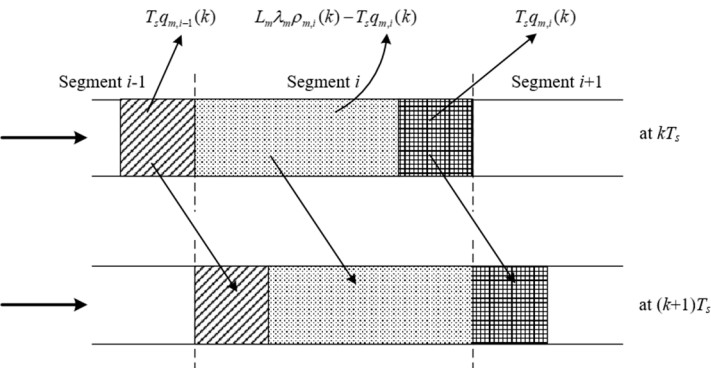

**Figure 2.** Temporal and spatial traffic flow in METANET.

The instantaneous speed of segment $i$ at $kT_s$ in the VT-micro model can be replaced by average speed $v_{m,i}(k)$ in the METANET model. $a^t_{m,i}(k)$ is defined as the temporal acceleration in the segment $i$ of link $m$ at time $kT_s$, while $a^s_{m,i,i+1}(k)$ is spatial acceleration of the vehicles leaving segment $i$ to segment $i+1$ of a link $m$. The corresponding number of vehicles is $n^t_{m,i}(k)$ and $n^s_{m,i,i+1}(k)$. As for the on-ramp,

its acceleration and number of vehicles are marked as $a_{on,o}$ and $n_{on,o}$. As shown in Figure 2. Following equations calculate the accelerations and the number of vehicles:

$$a^t_{m,i}(k) = \frac{v_{m,i}(k+1) - v_{m,i}(k)}{T_s} \tag{10}$$

$$a^s_{m,i,i+1}(k) = \frac{v_{m,i+1}(k+1) - v_{m,i}(k)}{T_s} \tag{11}$$

$$n^t_{m,i}(k) = L_m \lambda_m \rho_{m,i}(k) - T_s q_{m,i}(k) \tag{12}$$

$$n^s_{m,i,i+1}(k) = T_s q_{m,i}(k) \tag{13}$$

$$a_{on,o}(k) = [v_{m,i}(k+1) - v_{on,o}(k)]/T_s \tag{14}$$

$$n_{on,o}(k) = T_s q_{on,o}(k), \tag{15}$$

where $v_{on,o}$ denotes the average speed of the on-ramp $o$, and $q_{on,o}$ is the outflow of $o$. It is noted that there is no $v_{on,o}$ in the METANET model. Therefore, considering the regularity of traffic at peak time, this paper will estimate spatial acceleration of on-ramp based on history data [1]. Following equations calculate the emissions and fuel consumption with temporal acceleration:

$$J^t_{y,m,i}(k) = T_s n^t_{m,i}(k) \exp(V_{m,i}(k) P_y A^t_{m,i}(k)) \tag{16}$$

$$V_{m,i}(k) = [\ 1 \quad v_{m,i}(k) \quad (v_{m,i}(k))^2 \quad (v_{m,i}(k))^3\ ] \tag{17}$$

$$A^t_{m,i}(k) = [\ 1 \quad a^t_{m,i}(k) \quad (a^t_{m,i}(k))^2 \quad (a^t_{m,i}(k))^3\ ]^{\mathrm{T}}, \tag{18}$$

where $y \in \{CO, HC, NO_x, Fuel\ consumption\}$. $P_y$ is parameter matrix indexed from [1]. In this paper, the model proposed in [1] is used to consider the environmental effect, in which these four emission and fuel consumption factors are considered.

VT-micro emission model does not mention the estimated value of $CO_2$ emission rate. However, the authors of [1] show that there is an almost affine relationship between fuel consumption and $CO_2$ emission. Then $CO_2$ emission can be computed as follows:

$$J_{\alpha,CO_2}(l) = \delta_1 v_\alpha(l) + \delta_2 J_{\alpha,fuel}(l), \tag{19}$$

where $J_{\alpha,CO_2}(l)$ and $J_{\alpha,fuel}(l)$ denote the $CO_2$ emission rate and fuel consumption rate of vehicle $\alpha$ for time step $l$ respectively, $\delta_1$ and $\delta_2$ are model parameters. Although $CO_2$ is related to fuel consumption [23], the model considered in this paper is unfit for considering the environmental effect on the prospect of $CO_2$. It may take many changes by considering it in the model, therefore, we will only address this in future study.

It is the same to calculate emissions and fuel consumption with other two accelerations, named as $J^s_{y,m,i}(k)$ and $J_{on,o}(k)$. Thus, total emissions for various gases or fuel consumption at $kT_s$ of the freeway $m$ is described as follows:

$$J_y = \sum_k \sum_m \sum_i (J^t_{y,m,i}(k) + J^s_{y,m,i}(k)) + \sum_k \sum_{o \in O_{ramp}} J_{on,o}(k) \tag{20}$$

## 4. The Proposed Algorithm: MPC_CPDMO-NSGA-II

### 4.1. Framework of MPC_CPDMO-NSGA-II

In MPC, the controller is required to offer a solution based on current traffic conditions. Thus, the essence of freeway control problem is to realize on-line dynamic multi-objective optimization in model predictive control at each control step. Thus, dynamic multi-objective optimization is used to

solve the model predictive control problem to respond to changed environment rapidly, and to improve the performance of control solutions. In this paper, a new model predictive control method based on dynamic multi-objective optimization algorithms, named as MPC_CPDMO-NSGA-II is proposed to solve freeway control problem. Figure 3 shows the framework of the proposed MPC_CPDMO-NSGA-II algorithm. The difference between the MPC_CPDMO-NSGA-II algorithm and traditional MPC method is that it introduces dynamic multi-objective optimization algorithm instead of single objective optimization algorithm in MPC controller design.

Firstly, MPC_CPDMO-NSGA-II are used to provide more effective optimization solutions by dealing with multiple conflict objectives. Besides, it is more flexible to process constraints in the optimization period, such as being modified to new objectives optimized. It is easier to be realized in complex problems than linear relaxation method. Finally, MPC_CPDMO-NSGA-II can rapidly respond to the variation of traffic flow. For freeway congestion problems, environmental changes typically mean disturbance, weather conditions, traffic flow variations, etc. In this paper, in the disturbance, weather condition is not considered in the model, as the traffic flow is given based on on-site history data. In order to deal with the variation of traffic flow, the dynamic multi-objective genetic algorithm is used to solve the multi-objective optimization control problem in the model predictive control. Compared with static multi-objective genetic algorithm, dynamic multi-objective genetic algorithm can respond to the variation of traffic flow, i.e., environmental changes occur quickly and provide optimal solutions effectively.

In the MPC_CPDMO-NSGA-II algorithm, the dynamic multi-objective optimization algorithm will search for new optimal solutions by considering processed historical solutions as the initial population. If current optimal solutions are similar to the historical one, algorithms will perform a quick convergence. In contrast to static optimization algorithms, the initial population need to be re-generated randomly. Therefore, static optimization algorithms have lower efficiency and it is easy to acquire inaccurate solutions. In addition, MPC_CPDMO-NSGA-II is universal for application to solve various problems of optimization in the transportation field.

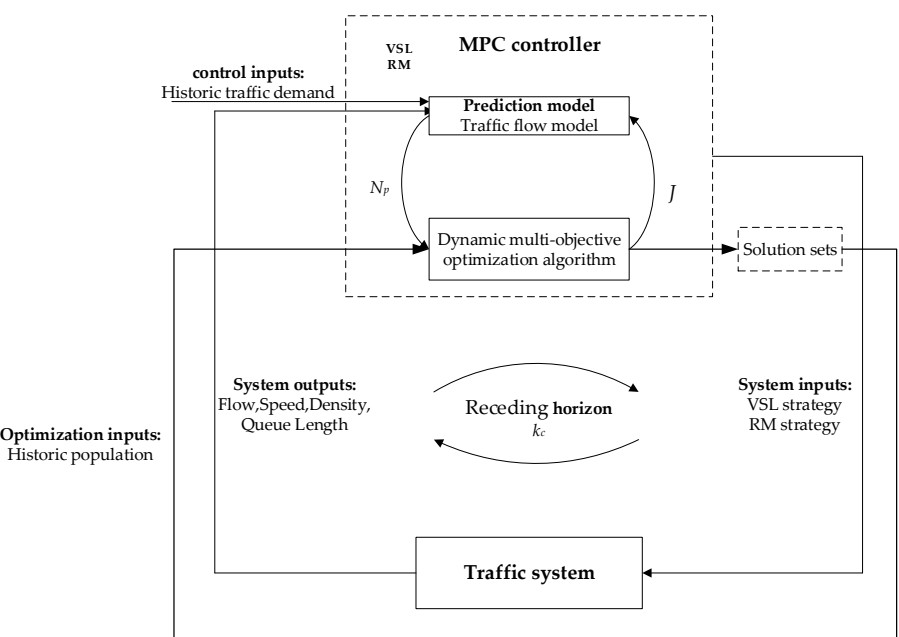

**Figure 3.** Framework of MPC_CPDMO-NSGA-II algorithm.

### 4.2. CPDMO-NSGA-II

In this paper, a new dynamic multi-objective optimization algorithm based on clustering prediction model, named as CPDMO-NSGA-II, will be proposed first, then the environmental detection method and prediction strategy used in CPDMO-NSGA-II algorithm will be discussed later.

The CPDMO-NSGA-II algorithm is used to realize on-line optimization in model predictive control at each control step.

The proposed CPDMO-NSGA-II algorithm is an improvement on the clustering prediction model based dynamic multi-objective evolutionary algorithm (CPM_DMOEA) proposed in paper [20]. In order to ensure a preferable distribution performance of the Pareto front, this paper first introduces a notion of reference points [24]. Reference points can describe the distribution of true Pareto front in several directions. They are used for determining the historical individuals on time series, which can alleviate the effects on algorithm's performance resulted from the poor distribution of individuals on time series, or time series decided by objectives values [19]. However, if true Pareto front is unknown, the choice of reference point loses its reference. Therefore, the static points are replaced with dynamic reference lines in this paper. Dynamic reference lines link static points with historical individuals, which enhance the distribution performance and population diversity simultaneously.

As for clustering prediction, the centers of clustering need to be predicted. The centers of clustering can describe current Pareto front obtained by dynamic multi-objective optimization algorithm, so it is reasonable to adopt reference points to determine the historical centers on time series. Besides, there are two more shortcomings in CPM_DMOEA in paper [20]: (1) The prediction model requires too many historical individuals, which occupies much storage space; and (2) Gaussian mutation used in shape prediction has greatly reduced population diversity. Therefore, in this paper, the VAR method, and the PRE method proposed by the authors of [18], are combined to predict the center of clustering. Furthermore, Gaussian mutation is also replaced with the mutation operator in CPM_DMOEA [20].

### 4.2.1. Description of CPDMO-NSGA-II

Figure 4 gives the flowchart of the proposed CPDMO-NSGA-II algorithm. We will describe the proposed CPDMO-NSGA-II algorithm firstly, then sub-algorithm 1 and sub-algorithm 2 used in the CPDMO-NSGA-II algorithm will be discussed later, finally, the environmental detection strategy and the prediction strategy used in the CPDMO-NSGA-II algorithm will be discussed.

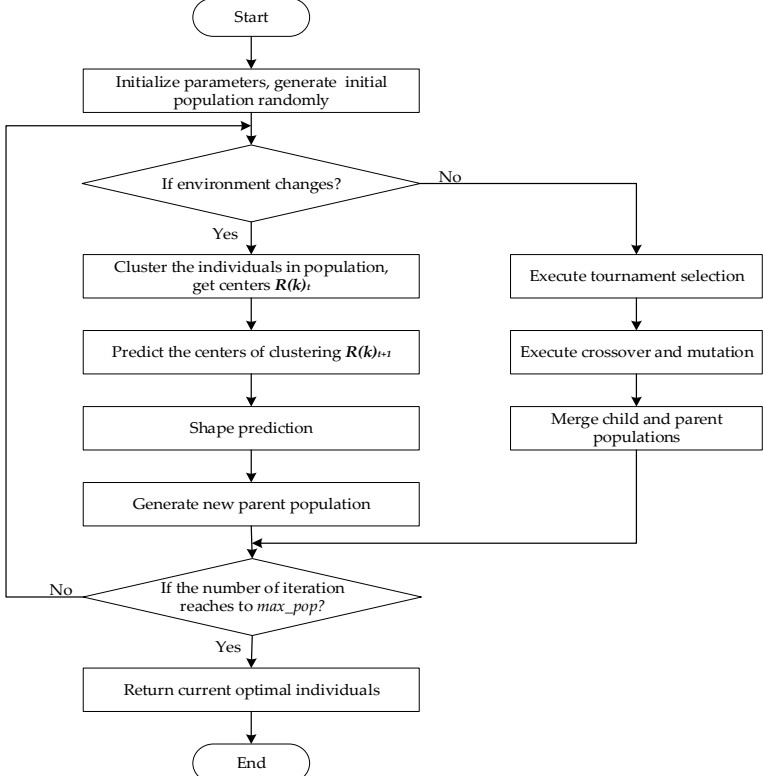

**Figure 4.** Flowchart of the CPDMO-NSGA-II algorithm.

(1)　The CPDMO-NSGA-II algorithm

Step 1: Initialize parameters and population: set maximal iteration max_*pop*, population size *pop*, initial time window $t = 1$, and generate initial population randomly;

Step 2: If environment changes, namely the value of detect operator exceeds threshold $\zeta$, go to the sub-algorithm 1 to realize clustering prediction, otherwise, go to step3.

Step 2.1: Set $Q_{t-1} = Q_t$, $Q_t = P_t$, For population $P_t$:

Reject $n$ individuals (the proportion is 25% in this paper), and generate $n$ new individuals $P_{random}$ randomly;

If $t < 3$, remain rest individuals, then go to Step2.3; if not, go to Step2.2, and generate $pop - n$ individuals through clustering prediction;

Step2.2: Clustering prediction:

Step 2.2.1 Cluster the individuals in population $P_t$ (execute sub-algorithm 1): for population $P_t$, getting centers $R(k)_t$, where $k = 1, 2, \ldots, cnum$, *cnum* is the number of clusters;

Step 2.2.2 Predict the centers of clustering (execute sub-algorithm 2): make association between $R(k)_t$ and $R(k)_{t-1}$ through algorithm 2, and predict *cnum* centers $R(k)_{t+1}$ through Equation (21);

Step 2.2.3 Execute shape prediction: predict the shape of each cluster according to $R(k)_{t+1}$, and generate $pop - n$ individuals $P_{predict}$. The operator of shape prediction is $x(k)_{t+1}^i = R(k)_{t+1}^i + \sigma$, where $\sigma$ is mutation operator in NSGA-II, $i = 1, 2, \ldots, n$, $n$ is the dimension of decision vector;

Step 2.2.4 Merge $P_{random}$ and $P_{predict}$, $P_{t+1} = P_{random} \cup P_{predict}$, as new parent population;

Step2.3: Go to next time window $t = t + 1$;

Step 3: Go to the process of evolution and selection;

Step3.1: Execute tournament selection, go to the process of evolution:

Step 3.1.1 Execute tournament selection for population $P_t$, and select $pop/2$ individuals to form a pool for crossover;

Step 3.1.2 Execute crossover and mutation for population $P_t$, and generate child population $C_t$;

Step 3.2: Merge child and parent populations, $P_t^{combine} = C_t \cup P_t$;

Step 3.2.1 Execute non-dominated sorting for $P_t^{combine}$, and calculation crowd distance;

Step 3.2.2 Select $pop$ optimal individuals through the selection operator in NSGA-II, as new parent population $P_{t+1}$;

Step 4: If the number of iterations reach max_*pop*, return current optimal individuals, then stop algorithm; if not, repeat Step 2 to Step 3.

In the proposed CPDMO-NSGA-II algorithm, the sub-algorithm 1 realized in Step 2.2.1, cluster the individuals in the population is used to realize how to cluster the individuals in population and get centers $R(k)_t$, which can avoid the poor distribution of historical population resulting in poor distribution performance of the whole algorithm. The sub-algorithm 1 is described as following:

(2)　Sub-algorithm 1: Cluster the individuals in the population

Step 1: Initialize a link list $G$; save population $P_t$; and determine the number of cluster *cnum*. Initial clusters and centers in $G$ are set as individuals themselves;

Step 2: For each cluster in $G$, calculate Euclidean distance between any two centers of clusters, recorded it into distance matrix $D$: $D_{ij} = \sqrt{\sum\limits_{m=1}^{M} (x_i{}^m - x_j{}^m)^2}$, where $x_i, x_j \in P_t$, which represents the moving direction of the individual in Pareto optimal set, $i \neq j = 1, 2, \ldots, pop$, $m = 1, 2, \ldots, M$, $m$ is the dimension of decision vector; *pop* is the size of $P_t$;

Step 3: Merge two clusters having minimal distance; update centers $R(k)_t^m = \frac{1}{|G[k]_t^m|} \sum\limits_{x_t^m \in G[k]_t^m} x_t^m$, where $k = 1, 2, \ldots, cnum$; and update distance matrix $D$;

Step 4: If the number of clusters in $G$ exceeds *cnum*, then repeat Step2–Step3.

In the proposed CPDMO-NSGA-II algorithm, the sub-algorithm 2 realized in Step 2.2.2, associate the individuals in the population is used to realize how to associate the individuals in the

population and get reference lines $RL_t$, which consider the distribution performance and population diversity simultaneously. The sub-algorithm 2 is described as following:

(3) Sub-algorithm 2: Associate the individuals in the population

Step 1: Place the individuals of population $P_t$ and reference point set $R$ in rectangular coordinates, expressed as vector $x_t^i$, $R_j$, where $i = 1, 2, \ldots, pop$, $j = 1, 2, \ldots, n$, $pop$ is the size of population $P_t$, and $n$ is the number of reference points, $x_t^i$ represents the *ith* vector predicted at time $t$;

Step 2: If environment first changes, initialize reference lines $RL_o = \left\{ x_t^i, R_j \right\}$, then go to Step 4;

Step 2.1: For each individual $x_t^i$ in $P_t$, calculate the cosine of angles between $x_t^i$ and each reference point $R_j$: $\cos \alpha_{ij} = \dfrac{x_t^i \cdot R_j}{|x_t^i| \cdot |R_j|}$;

Step 2.2: Connect $x_t^i$ and $R_j$ having minimal angles $\alpha_{ij}$ (or $\cos \alpha_{ij}$ is maximal, see Figure 5a): $RL_t^k = \left\{ x_t^i, R_x^j \right\}$, where $k = 0, 1, \ldots, K$, $K$ is the number of reference lines;

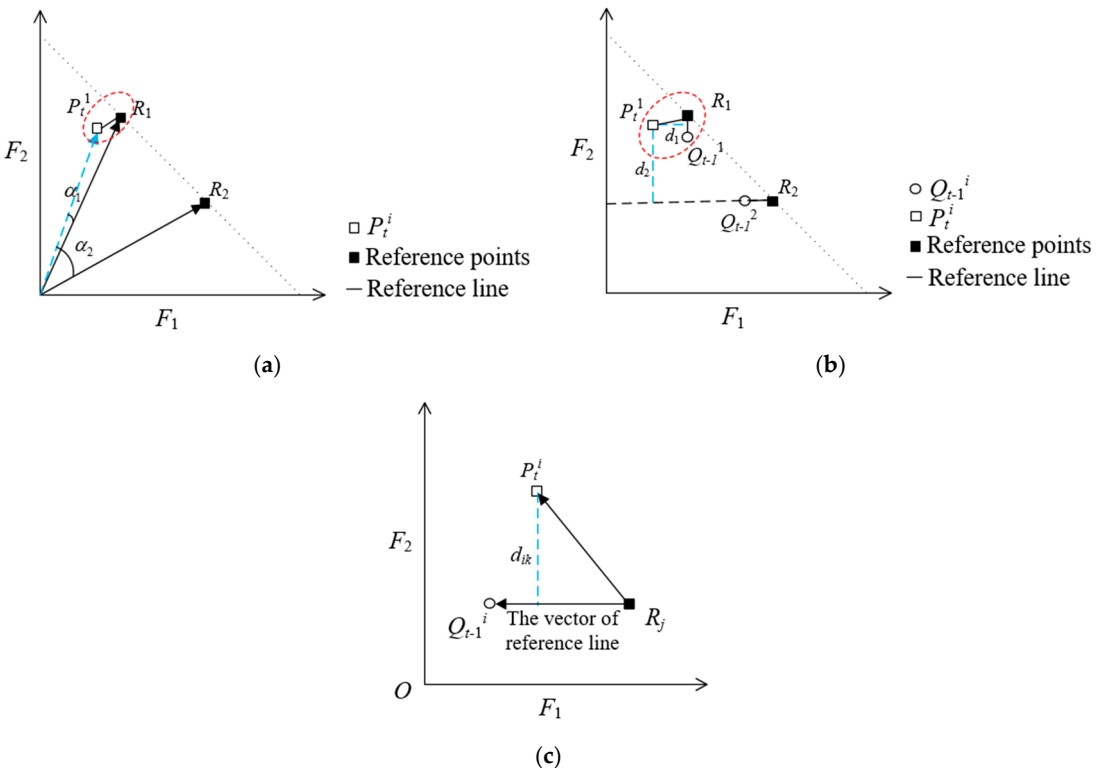

(a)

(b)

(c)

**Figure 5.** Relation determination of individuals between time $t$ and $t-1$. (**a**) Relation determination by angles; (**b**) Relation determination by distances; (**c**) The distance between a point and vector.

Step 3: If not, update reference lines merge $RL_t = \{x_t, R_x\}$, then go to Step 4;

Step 3.1: For each individual $x_t^i$ in $P_t$, calculate the distances $d_{ik}$ between $x_t^i$ and each current reference point $RL_t^k$: i.e., $d_{ik} = \left| \overrightarrow{R_j P_t}^i - \overrightarrow{R_j Q_{t-1}}^i \cdot \dfrac{(\overrightarrow{R_j P_t}^i \cdot \overrightarrow{R_j Q_{t-1}}^i)}{\left| \overrightarrow{R_j Q_{t-1}}^i \right|^2} \right|$.

Step 3.2: Connect $x_t^i$ and corresponding reference point having minimal distance (see Figure 5b), and $x_t^i$ is associated with historical individual $Q_{t-1}$ on historical reference line $RL_{t-1}$. (see Figure 5c)

Step 4: Save reference lines $RL_t = \{x_t, R_x\}$.

In general, reference points are uniformly distributed on a hyperplane. Reference point set $R$ is set as follows [25]:

$$R : \begin{cases} R_j = \left\{ R_j^1, R_j^2, \ldots, R_j^m \right\} \\ R_j^m \in \{0/p, 1/p, \ldots, p/p\}, \displaystyle\sum_{m=1}^{M} R_j^m = 1 \end{cases},$$

where $j = 1, 2, \ldots, n$, $n$ is the number of reference points, $n = \binom{M + p - 1}{p}$; $m = 1, 2, \ldots, M$, $M$ is the dimension of reference points, $p = M$. Normally, set $n \approx pop$, $pop$ is the size of population. Figure 5 shows the relation determination method of individuals between time $t$ and $t - 1$. The concrete method is described in association algorithm for historical individuals. Figure 5a realized in Step 2.2 in sub-algorithm2 shows the relation determination by angles, Figure 5b realized in step 3.2 in sub-algorithm2 shows the Relation determination by distances, Figure 5c realized in Step 3.2 in sub-algorithm2 shows the distance between a point and vector. $F_i$ is just a coordinate axis and it doesn't mean anything, $P_t^i$ represents the *ith* Pareto optimal solution at time $t$ in the rectangular coordinate system, $Q_{t-1}^i$ represents the *ith* Pareto optimal solution at time $t - 1$ in the rectangular coordinate system.

### 4.2.2. Environmental Detection

Now, we will discuss the environmental detection strategy realized in Step 2 in the proposed CPDMO-NSGA-II algorithm. In general, similarity test is a common strategy for detecting environmental changes. The basic idea is to reevaluate objectives or constraints based on the individuals selected randomly from current population. To some extent, the environment is considered as change if the value of objectives or constraints has changed. In this paper, the detection operator proposed in paper [19] is used as follows:

$$\varepsilon(t) = \frac{\sum\limits_{i=1}^{n_\varepsilon} \| f(x^i, t) - f(x^i, t-1) \|}{n_\varepsilon \max\limits_{i=1,\ldots,n_\varepsilon} \| f(x^i, t) - f(x^i, t-1) \|} \tag{21}$$

Suppose that there are $n_\varepsilon$ individuals selected to detect change, and $x^i$ is the *ith* individual. $f(x^i, t-1)$ is the objective at $x^i$ at time $t - 1$, while $f(x^i, t)$ is the objective at time $t$. If $\varepsilon(\tau) \geq \eta$, where $\eta$ is a threshold value, it can be acknowledged as environmental change. For the sake of effectively searching optimal solutions in new environment, the algorithm should enhance the population diversity through special methods.

### 4.2.3. Prediction Strategy

Now, we will discuss the prediction strategy realized in Step2 in the proposed CPDMO-NSGA-II algorithm. When a change is detected, prediction strategy contributes to a faster convergence to new Pareto front. In this paper, the VAR method and PRE method proposed in paper [18] are combined to employ prediction, named as V&P method. Half of the initial population at time $t + 1$ is distributed around predicted individuals, and the rest are generated by current population. Following is the formula of V&P method:

$$x_{t+1} = \begin{cases} x_t + (x_t - x_{t-1}) + \eta, \ rand() < 0.5; \\ x_t + \eta, \qquad otherwise. \end{cases} \tag{22}$$

where $rand()$ returns a random number within lower bound 0 and upper bound 1. Gaussian noise $\eta$ is defined by:

$$\eta \sim N(0, \delta I), \tag{23}$$

where $I$ is an unit matrix, and $\delta$ is standard deviation, which is defined as:

$$\delta^2 = \frac{1}{4n} \|x_t - x_{t-1}\|_2^2, \tag{24}$$

where $n$ is the number of decision vector. Generally, in PRE method, Equation (24) is used to find historical individual $x_{t-1}$ with the same convergence direction as $x_t$.

$$x_{t-1} = \arg\min_{y \in Q_{t-1}} \|y - x_t\|_2. \tag{25}$$

## 5. Problem Formulation

This section mainly explains the optimization objectives in MPC_CPDMO-NSGA-II. This paper adopts TTS, TTD, TE and TF to evaluate the performance of the freeway, which are also used as the optimization objectives in dynamic multi-objective optimization algorithm CPDMO-NSGA-II. The prediction models for estimating these indicators are the integration of METANET and VT-micro model.

(1) Total Time Spend (TTS)

The TTS consists of two parts: (1) Total travel time (TTT); (2) Total waiting time (TWT).

$$\min \quad J_1(k_c) = \sum_{k=Zk_c}^{Z(k_c+N_p)-1} T_s \cdot \left[ \sum_m \sum_i \rho_{m,i}(k) \cdot L_{m,i} \cdot \lambda_{m,i} + \sum_{o \in Ramp} \omega_o(k) \right], \tag{26}$$

where $k$ is steps of sampling; $k_c$ is steps of control; $N_p$ is prediction horizon; and $Z$ is the ratio of control period and sampling period.

(2) Total Travel Distance (TTD)

$$\max \quad J_2(k_c) = \sum_{k=Zk_c}^{Z(k_c+N_p)-1} \sum_m q_{m,i} \cdot T_s . \tag{27}$$

In optimization, maximized objectives will be transformed into minimized problems with a minus.

(3) Total Emissions (TE) and Total Fuel Consumption (TF)

According to the VT-micro model, following equation is used to calculate the TE for various gases and the TF:

$$J_y(k_c) = \sum_{k=Zk_c}^{Z(k_c+N_p)-1} \left[ \sum_{i=1}^{3} (J^t_{y,m,i}(k) + J^s_{y,m,i}(k)) + J_{y,on,o}(k) \right], \tag{28}$$

where $y \in \{CO, HC, NO_x\}$. It should be noted that TE and TF are two different indicators, so they cannot be added together directly. In this paper, the problem is solved by standardization using the following:

$$\min \quad J_3(k_c) = \frac{J_{CO}(k_c) + J_{HC}(k_c) + J_{NO_x}(k_c)}{J_{sl,emisson}(k_c)} + \frac{J_{FC}(k_c)}{J_{sl,FC}(k_c)}, \tag{29}$$

where $J_{sl,emisson}(k_c)$ is TE of the freeway with fixed speed limit strategy, and $J_{sl,FC}(k_c)$ is TF. They can be obtained through the follow-up simulation experiment.

To ensure the safety of the environment, the control solutions should not have too much fluctuation. Besides, the flows of the freeway should not exceed capacity. Therefore, following constraints are considered in the optimization.

(1) In the same control horizon, the difference of VSLs between adjacent segments is no more than 5 km/h:

$$\left| v_{ctrl,i}(k_c) - v_{ctrl,j}(k_c) \right| \leq 5 \tag{30}$$

(2) At the same segment, the difference of VSLs between adjacent control horizons is no than 10 km/h:

$$\left| v_{ctrl,i}(k_c) - v_{ctrl,i}(k_c + 1) \right| \leq 10 \tag{31}$$

(3) Between adjacent control horizons, the difference of RM rates is no more than 0.3:

$$\left| r_o(k_c) - r_o(k_c + 1) \right| \leq 0.3 \tag{32}$$

(4)  The range of RM rate is shown as follows:

$$r_{o,\min} \le r_o(k_c) \le 1 \tag{33}$$

(5)  The flows of the mainline and the on-ramp should not exceed the traffic capacity:

$$q_{m,i}(k) - Q_{main,\max,i} \le 0 \tag{34}$$

$$q_o(k) - Q_{o,\max} \le 0, \tag{35}$$

where $v_{ctrl,i}(k_c)$ is the value of VSL applied to segment $i$ at control horizon $k_c$; $Q_{main,\max,i}$ is the capacity of segment $i$; $Q_{o,\max}$ is the RM rate applied to the on-ramp $o$ at control horizon $k_c$; and $Q_{o,\max}$ is the capacity of the on-ramp $o$. Some parameters are haven been described in Section 3. Above constraints are referred to paper [26,27]. Due to a shorter control period in this paper, some of them are modified in order to alleviate risk of slow response from drivers to changed control solutions. This paper employs static penalty function to dispose these constraints, so that the CPDMO-NSGA-II algorithm can search for non-dominated solutions in feasible domains. These penalties are added to existed objectives:

$$\min \quad J_1(k_c) = J_1(k_c) + \phi_1 \Big( \sum_{k=Zk_c}^{Z(k_c+N_p)-1} \sum_i \max(0, q_{m,i}(k) - Q_{main,\max,i}) + \sum_{k=Zk_c}^{Z(k_c+N_p)-1} \max(0, q_o(k) - Q_o)) \tag{36}$$

$$\min \quad J_3(k_c) = J_3(k_c) + \phi_2 \Big( \sum_j \sum_{k=Zk_c}^{Z(k_c+N_c)-1} \max(0, |v_{ctrl,1}(k) - v_{ctrl,2}(k)| - 5) + \sum_j \sum_{k=Zk_c}^{Z(k_c+N_c)-2} \max(0, |v_{ctrl,j}(k) - v_{ctrl,j}(k+1)| - 10) + \sum_{k=Zk_c}^{Z(k_c+N_c)-2} \max(0, |r_o(k) - r_o(k+1)| - 0.3)) \tag{37}$$

where $j$ is the number of the segment applied VSL, and $N_c$ is control horizon. In this article, the VSL strategy is executed on segment 1 and 2. we will explain the reason in Section 5. It is found that the performance of the CPDMO-NSGA-II algorithm is better with penalty coefficients 0.029 and 0.2, determined by trial and error experiment.

In the process of receding optimization, it will provide a set of solutions that are non-dominated. But only one optimal solution can be executed in the highway considering the preference of decision-makers. In essence, it is a process of multi-attribute decision-making. TOPSIS is one such efficient method [21]. The weights of TOPSIS are determined subjectively. The efficiency of freeway expressed by TTS is the most important, closely followed by TTD and the sum of emissions and fuel consumption, so the corresponding of both model predictive control based on single-objective optimization algorithm (MPC_SOO) and MPC_CPDMO-NSGA-II weights are 0.4, 0.3, 0.3. The influence of different weights on experimental results can be discussed in the future.

## 6. Simulation Research

The MPC_CPDMO-NSGA-II algorithm proposed in Section 4 is used to solve the freeway congestion control problem with an on-ramp during rush hour.

### 6.1. Simulation Network

Figure 6 shows the highway with an on-ramp in simulation study. The length of main road is 1500 m. It has three lanes in the mainline, but one lane in the on-ramp. In this paper, the METANET model is applied to simulate behaviors of the highway. Thus, the highway is divided into three

segments, each is 500 m. The on-ramp is located in the second segment. The parameters of the METANET model is referred to paper [7] (see Table 1). Figure 7 is the traffic demand of the mainline and the on-ramp. It regenerates a peak hour. In this scenario, the capacity of the mainline is 2000 veh/km/lane, and the on-ramp is 1500 veh/km/lane.

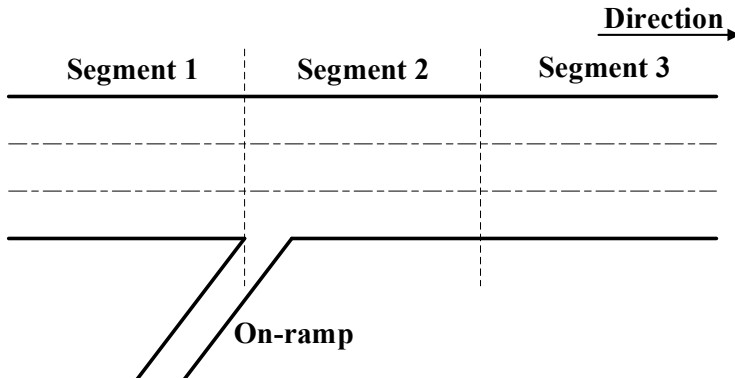

**Figure 6.** The freeway network with an on-ramp.

**Table 1.** Parameters of the METANET model.

| Name | Value | Unit | Name | Value | Unit |
|---|---|---|---|---|---|
| $\tau$ | 18 | s | $\rho_{cr,m}$ | 33.5 | veh/km/lane |
| $\kappa$ | 40 | veh/km/lane | $\delta$ | 0.012 | - |
| $\upsilon$ | 60 | km$^2$/h | $\alpha_m$ | 1.636 | - |
| $\rho_{max}$ | 180 | veh/km/lane | $v_{f,m}$ | 110 | km/h |

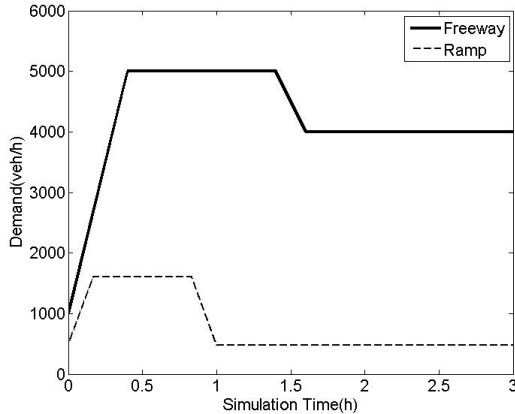

**Figure 7.** The traffic demands of the mainline and the on-ramp.

As shown in Figure 7, the initial demands of the mainline and the on-ramp are small. At the beginning, they are in the trend of linear growth. After about 20 min, the demand of the mainline reaches to 5000 veh/h, while the on-ramp reaches to 1500 veh/h in about 10 min. Both of them remains constant in a period of time. Close to 1.5 h, the demand of the mainline begins to decline, then remains 4000 veh/h to the end. The demand of the on-ramp reduces at 50 min, then it arrives to 500 veh/h and keeps constant.

*6.2. Simulation Results*

It is assumed that all of vehicles are cars, and the compliance rate is 100%, namely $a = 0$. The parameters of MPC_CPDMO-NSGA-II are shown in Table 2, which are also referred by the authors of [7]. Table 2 gives the value of parameter use in the MPC_CPDMO-NSGA-II algorithm. It can be seen from Table 2 that the sampling period is set as 10 s, the control period is set as 1 min, the prediction

horizon and the control horizon are set as 15 min and 10 min respectively in this paper. Since the control period is set as 1 min, it means that the MPC_CPDMO-NSGA-II algorithm proposed in this paper is suitable for both the simulation experiment and on-line control on the small road network. For larger networks, since more computation time are required, other methods should be applied. The sampling period is also the prediction period of METANET model.

**Table 2.** Parameters of MPC_CPDMO-NSGA-II.

| Name | Value |
|---|---|
| **Simulation time** | 3 h |
| **Sampling period ($T_s$)** | 10 s |
| **Control period** | 1 min |
| **Prediction horizon** | 15 min |
| **Control horizon** | 10 min |

In this paper, we will analyze the conditions of the freeway with fixed speed limit strategy firstly, so as to determine which segment requires freeway control strategy. Then, the proposed MPC_CPDMO-NSGA-II I algorithm is used to solve freeway congestions in rush hours. The control strategy is an incorporation of VSL and RM strategy. In the proposed CPDMO-NSGA-II algorithm, the size of population is 160; the number of iterations is 50; and the number of clustering is 10.

For comparison, a model predictive control based on single-objective optimization algorithm named as MPC_SOO, is used to solve the freeway control problem in this paper, and the single-objective optimization algorithm is genetic algorithm [28]. The performance indicator used in MPC_SOO is described as following:

$$\min \quad J(k) = k_1 J_1(k) - k_2 J_2(k) + k_3 J_3(k), \tag{38}$$

where $k_1, k_2, k_3$ are the weights, which are set as 0.4, 0.3, 0.3 respectively. To compare the result of the MPC_SOO algorithm and the MPC_CPDMO-NSGA-II algorithm under similar expert's experience, the weights of the $k_1, k_2, k_3$ used in the MPC_SOO algorithm is set as the same value used in the TOPSIS method in the MPC_CPDMO-NSGA-II algorithm. In the MPC_SOO method, the single objective is set as the weight sum of the multiple objectives considered in the MPC_CPDMO-NSGA-II algorithm, and the weights used in the single objective in the MPC_SOO method is set as 0.4, 0.3, 0.3, which is the same value used in the TOPSIS method in the MPC_CPDMO-NSGA-II algorithm. The reason is that the result of the MPC_SOO method and the MPC_CPDMO-NSGA-II algorithm are compared under similar expert's experience. In order to distinctly analyze the trends of traffic variables, they are averaged before plotting every 2 min.

### 6.2.1. Results of Traffic Condition, Emissions and Fuel Consumption of the Freeway with Fixed Speed Limit

In the whole simulation process, the fixed speed limit is directly implemented in VISSIM simulation environment, and minimum and maximum speed of the vehicle on each segment on the road is set at [60,100] km/h.

In this part, we will discuss the results obtained from fixed speed limit strategy firstly. The Pareto fronts obtained by MPC_CPDMO-NSGA-II algorithm will be discussed in Section 6.2.2 secondly. The results obtained from MPC_SOO method and MPC_CPDMO-NSGA-II algorithm will be discussed in Section 6.2.3 thirdly. The results about traffic conditions obtained from fixed speed limit strategy, MPC_SOO method and MPC_CPDMO-NSGA-II algorithm will be discussed in Section 6.2.4.

Figure 8a–d gives the variation of the average flow, the average speed, the average density of the mainline and the queue length of the on-ramp and its upstream mainline obtained from fixed speed limit strategy. It can be seen from Figure 8a–c that the flow of segment 1 is always less than other two segments due to the entered vehicles from the on-ramp. With the increasing demand of the mainline, its flow and density also increase gradually, but the speed decreases. When the demand

remains at peak, the average flow, density and speed fluctuate around some value till the demand declines, the main stream traffic also will be reduced. At 1.5 h, the density begins to drop, and speed rises. They are in slight fluctuations to the end.

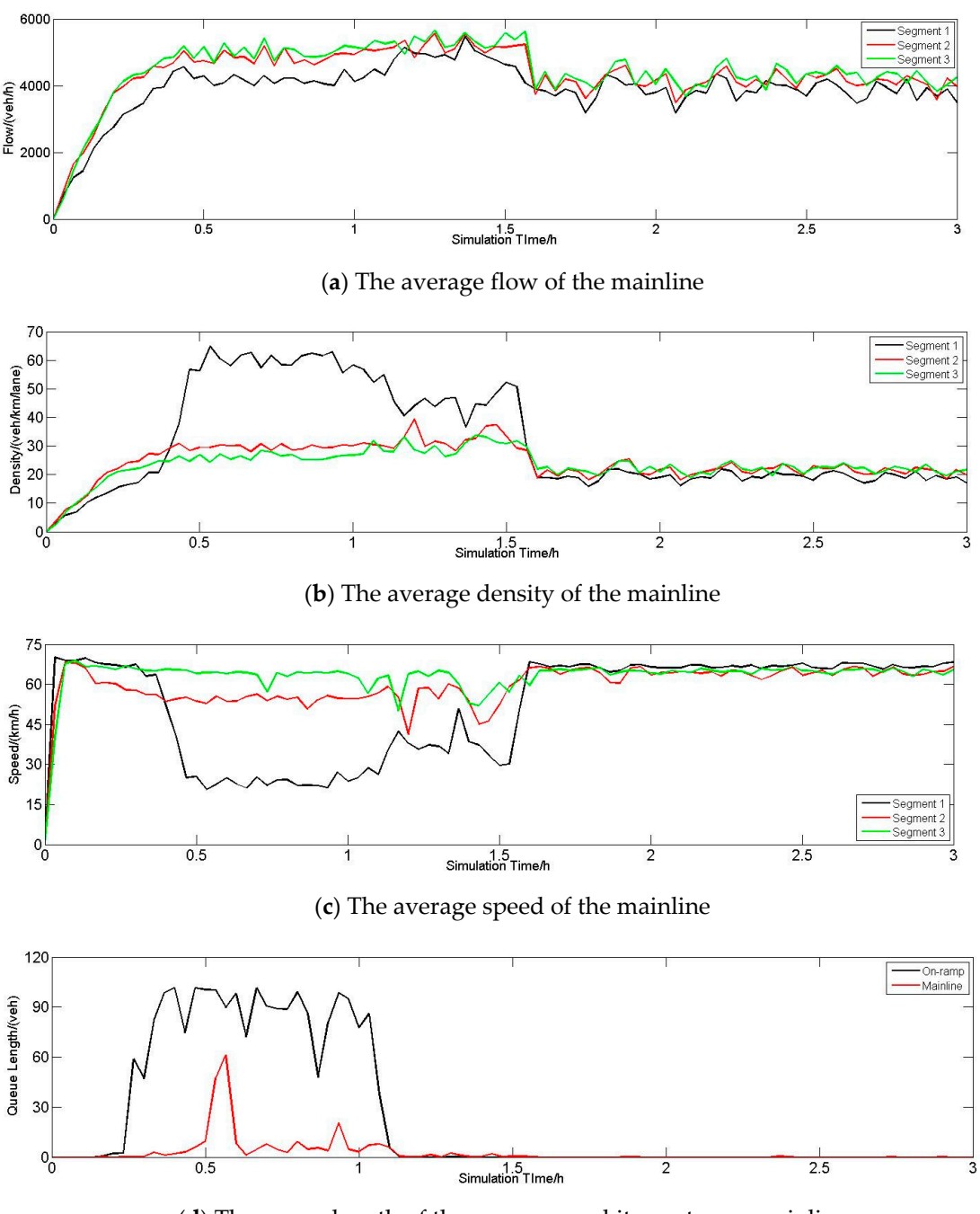

(**a**) The average flow of the mainline

(**b**) The average density of the mainline

(**c**) The average speed of the mainline

(**d**) The queue length of the on-ramp and its upstream mainline

**Figure 8.** The traffic conditions of the freeway with fixed speed limit.

It can be seen from Figure 8d that the queue length rises with the increasement of the demand on the on-ramp. The peak is more than 100 vehicles. After a severe congestion in an hour, the queue disappears rapidly, and there is no queue anymore. This article also samples the queue length of upstream mainline of the on-ramp. As shown in Figure 8d, its queue is shorter. The maximum is only about 60 vehicles. After 1 h 15 min, there is almost no waiting vehicles.

Figure 9 shows the trends of emission and fuel consumption of the freeway with fixed speed limit. In the period of increasing flow and forming congestion of the on-ramp and its upstream mainline,

emissions and fuel consumption rise. When there is a severe congestion of the on-ramp, they remain high values. Although the queues disappear, there is no downtrend until the vehicles of the mainline have higher speeds. After nearly two hours, emissions and fuel consumption fluctuate around 0.1 L and 5 kg respectively. In this way, we will execute VSL on segment 1 and 2 and RM on on-ramp in this paper.

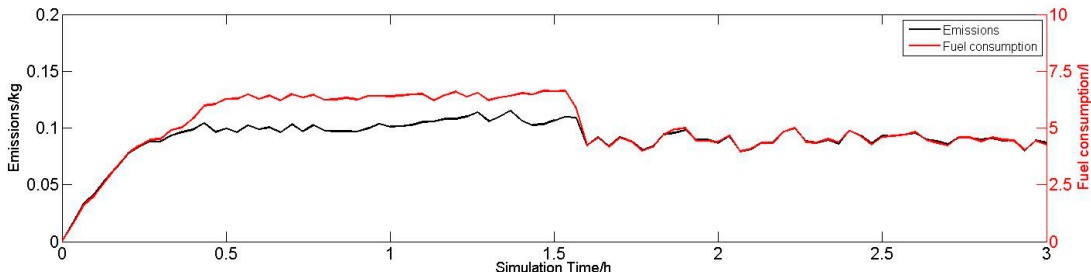

**Figure 9.** Emissions and fuel consumption of the freeway with fixed speed limit.

### 6.2.2. Pareto Fronts Obtained by MPC_CPDMO-NSGA-II Algorithm

Figure 10 give the Pareto fronts obtained by MPC_CPDMO-NSGA-II in a receding horizon. All of the objectives are normalized. It can be seen that there is no conflict between TTS and TTD, but they are obviously conflicting with sum of TE and TF respectively. The conflicts are similar.

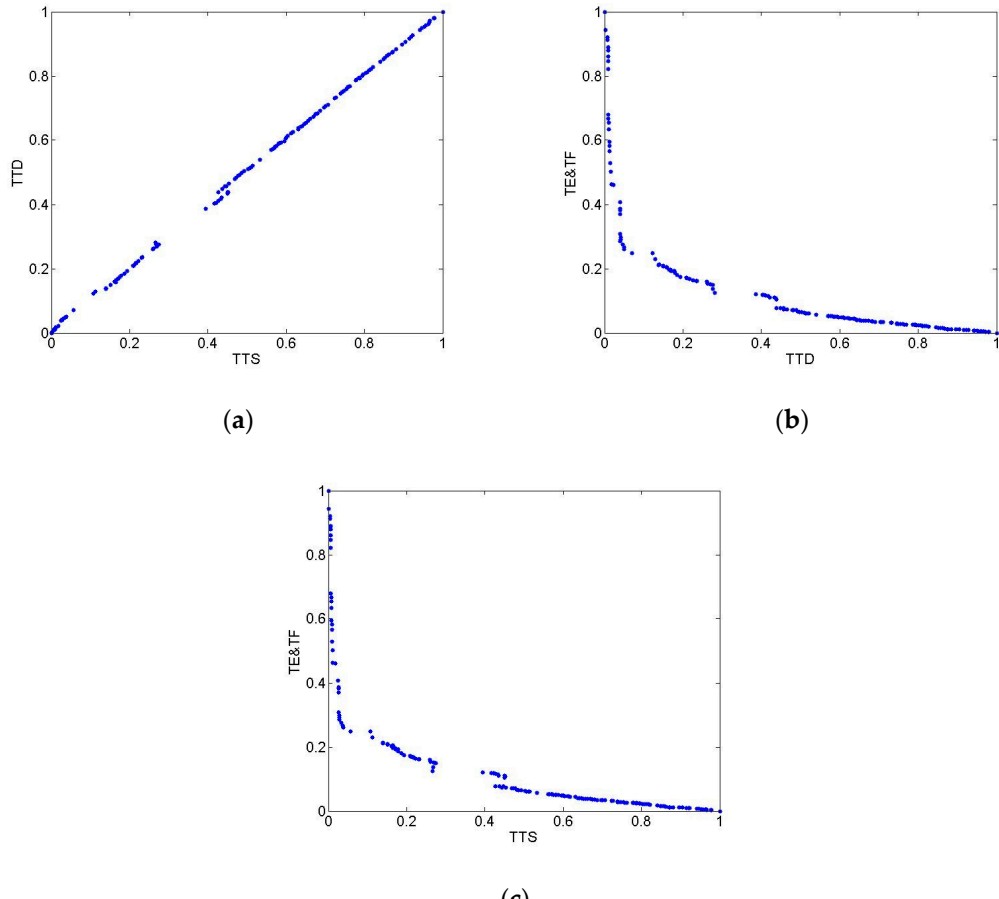

**Figure 10.** Pareto fronts in a horizon of receding optimization with MPC_CPDMO-NSGA-II. (**a**) The Pareto front of total time spend (TTS) and total travel distance (TTD); (**b**) The Pareto front of TTD, and TE and TF; (**c**) The Pareto front of TTS, and TE and TF.

Figure 11 displays the control solution with MPC_SOO and MPC_CPDMO-NSGA-II, including VSLs for segment 1 and 2, and RM strategy. In Figure 11, the actual road network is taken into account, only VSLs are considered for segment 1 and 2, and no VSLs are implemented in Section 3. However, we supplement the flow, density and speed of segment 3 under different methods in Table 4. In terms of safety, the MPC_CPDMO-NSGA-II offers a smoother solution that can reduce the risk of driver's slow respond to changes. From 10 min to 1 h, the value of VSLs are small with proposed method. Afterwards, it provides larger VSLs most of the time. As for ramp metering rate, it allows more vehicles to enter into the mainline than MPC_SOO most of the time.

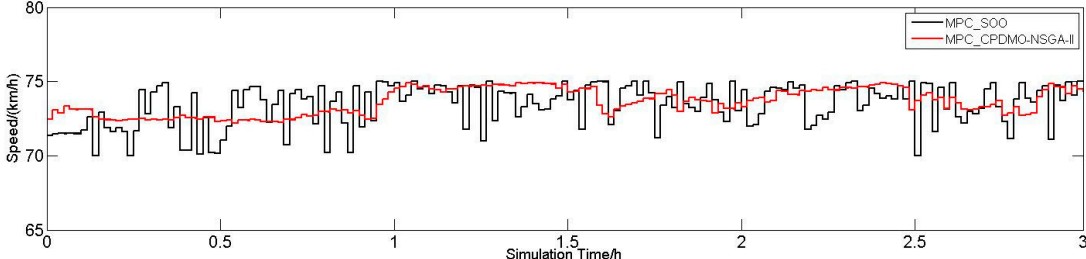

(**a**) Values of variable speed limit (VSL) of segment 1 with MPC_SOO and MPC_CPDMO-NSGA-II

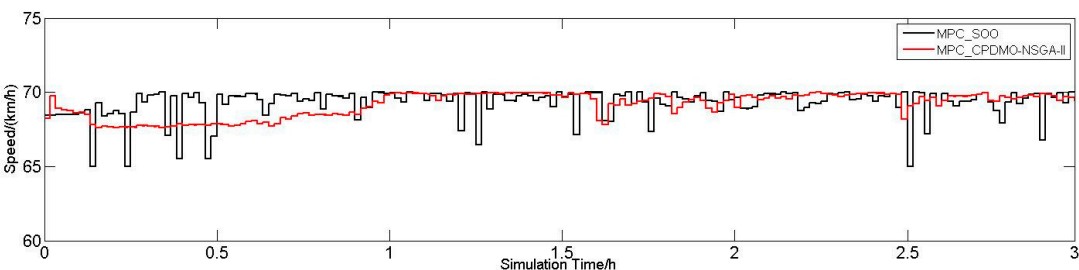

(**b**) Values of VSL of segment 2 with MPC_SOO and MPC_CPDMO-NSGA-II

(**c**) Values of RM of the on-ramp with MPC_SOO and MPC_CPDMO-NSGA-II

**Figure 11.** The control solutions of the freeway with MPC_SOO and MPC_CPDMO-NSGA-II.

### 6.2.3. Performance of the Freeway

In this paper, the performance of the freeway is evaluated from the aspect of TTS, TTD, TE and TF. Table 3 lists these indicators with three control methods, namely fixed speed limit, MPC_SOO and MPC_CPDMO-NSGA-II. For reducing the effects of random seeds, this paper repeats five experiments for each method, and calculates the mean value of the performance indicators and traffic variables.

**Table 3.** The performance of the freeway with three control methods.

| Indicators<br>Methods | TTS (h) | TTD (km) | TE (kg) | TF (l) |
|---|---|---|---|---|
| Fixed speed limit | 554.217 | 29,228.58 | 99.554 | 5493.556 |
| MPC_SOO | 465.8766(−15.9%) | 30,564.134.6% | 38.687(−61.1%) | 1775.608(−67.7%) |
| MPC_CPDMO-NSGA-II | 459.8024(−17%) | 30,653.244.9% | 38.576(−61.3%) | 1767.215(−67.8%) |

From Table 3, it is known that the efficiency of the freeway is the worst with fixed speed limit. It also aggravates the burdens of environment and economics. By contrast, MPC_SOO and MPC_CPDMO-NSGA-II greatly improve the performance of the freeway, especially in the aspect of TE and TF, reduced more than 60%. In MPC_SOO, TTS is 15.9% shorter than fixed speed limit, while TTD rises by 4.6%. It implies that vehicles passed the freeway increase, or their speed increases significantly. Compared to MPC_SOO, the presented method reduces more TTS, TE and TF, and greater improves TTD, reaching to 4.9%.

6.2.4. Discussion about Traffic Conditions

Then we will explore the effect of traffic variables, including average flow, density and speed, with three control methods in this paper. Since an on-ramp located in segment 2, bottleneck often occurs in that area, so we mainly analysis the variation of traffic variables on segment 2 (see Figure 12). Generally, MPC_SOO and MPC_CPDMO-NSGA-II perform better. They improve the situation of low efficiency resulted from fixed speed limit, proved by greatly enhanced flow and speed. From 1 h to 1.5 h, the density of segment 2 decreases a lot.

It can be seen from Figure 12 that the results of MPC_CPDMO-NSGA-II algorithm and MPC_SOO method are too similar, therefore the results of traffic conditions on segment 1, 2, 3 are discussed in Table 4. It can be concluded from Table 4 that compared to fixed speed limit strategy, the results of traffic conditions perform better under the MPC_SOO method and MPC_CPDMO-NSGA-II algorithm. For example, in the aspect of traffic flow, the average flow on segment 2 is increased by 7.4% and 7.9% respectively under the MPC_SOO method and MPC_CPDMO-NSGA-II algorithm compared with the fixed speed limit strategy, in the aspect of average density, it is decreased by 40.6% and 40.7% respectively on segment 2 under the MPC_SOO method and MPC_CPDMO-NSGA-II algorithm compared with the fixed speed limit strategy, in the aspect of average speed, it is increased by 14.0% and 14.2% respectively on segment 3 under the MPC_SOO method and MPC_CPDMO-NSGA-II algorithm compared with the fixed speed limit strategy.

Figure 13 displays the queue lengths of the on-ramp and its upstream mainline with three control methods. It can be seen from Figure 13 that in the period of high demand of the on-ramp, the RM adopted in MPC_SOO greatly reduces the queue length, and shortens time of severe congestion. It should be noted that the queue is formed naturally with fixed speed limit, while may be resulted actively by traffic lights with other two methods. Thus, in MPC_SOO, there are slight vehicles wating in the on-ramp after 1 h, indicating that it may put efficiency of the mainline first. However, the heavy congestion the on-ramp disappears by using MPC_CPDMO-NSGA-II, with a peak valued 11. Afterwards, there is almost no waiting vehicle of the on-ramp.

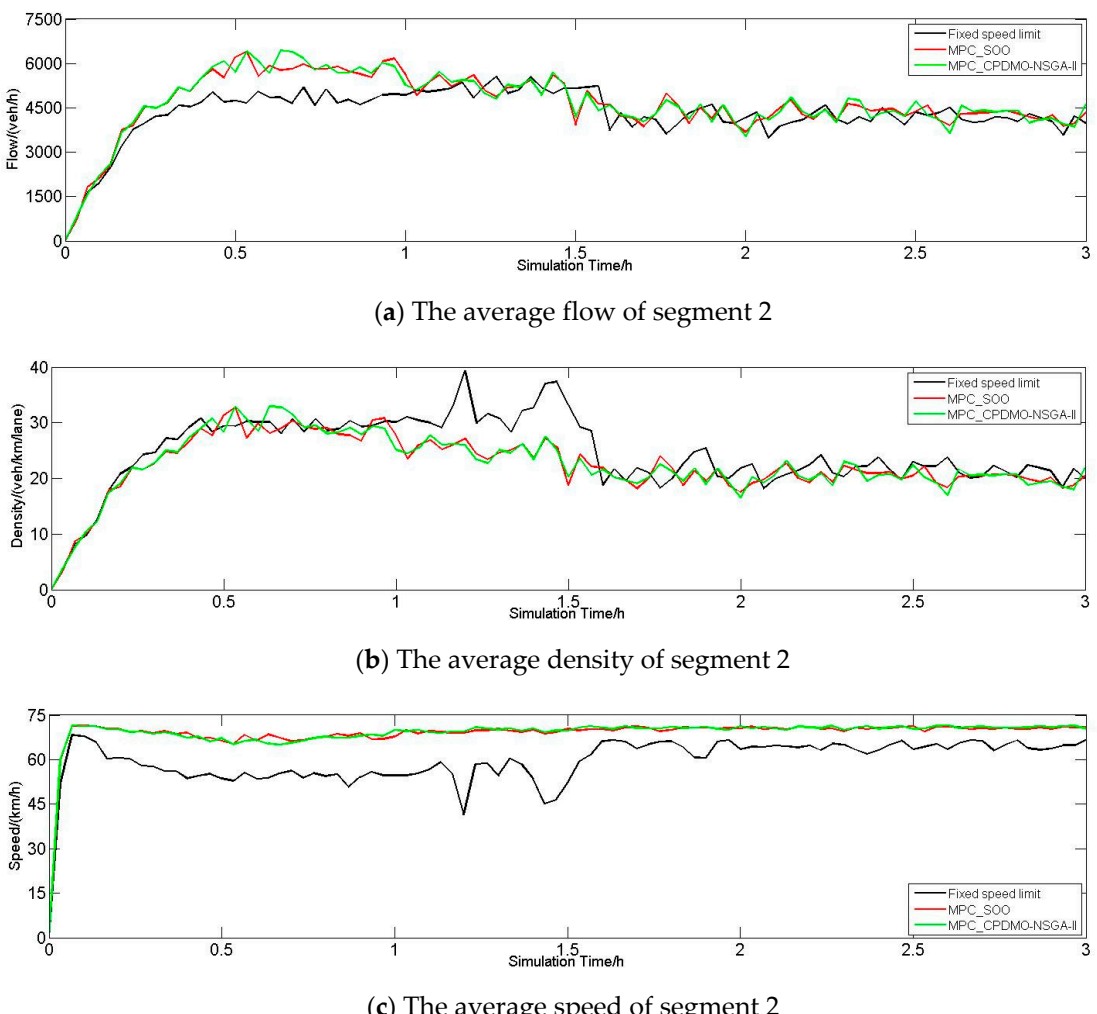

(**a**) The average flow of segment 2

(**b**) The average density of segment 2

(**c**) The average speed of segment 2

**Figure 12.** The traffic conditions of segment 2 using three control methods.

**Table 4.** The results of traffic conditions on segment 1, 2, 3 using three control methods.

| Traffic Conditions | Methods | Fixed Speed Limit | MPC_SOO | MPC_CPDMO-NSGA-II |
|---|---|---|---|---|
| **Flow (veh/h)** | Segment 1 | 3942.02 | 4124.11 4.6% | 4121.85 4.6% |
| | Segment 2 | 4360.05 | 4682.84 7.4% | 4704.81 7.9% |
| | Segment 3 | 4489.31 | 4652.83 3.6% | 4670.68 4.0% |
| **Density (veh/km/lane)** | Segment 1 | 31.55 | 18.73(−40.6%) | 18.72(−40.7%) |
| | Segment 2 | 24.70 | 22.56(−8.7%) | 22.62(−8.4%) |
| | Segment 3 | 23.60 | 21.35(−9.5%) | 21.40(−9.3%) |
| **Speed (km/h)** | Segment 1 | 52.37 | 73.49 40.3% | 73.52 40.4% |
| | Segment 2 | 60.07 | 69.55 15.8% | 69.63 15.9% |
| | Segment 3 | 63.67 | 72.62 14.0% | 72.70 14.2% |

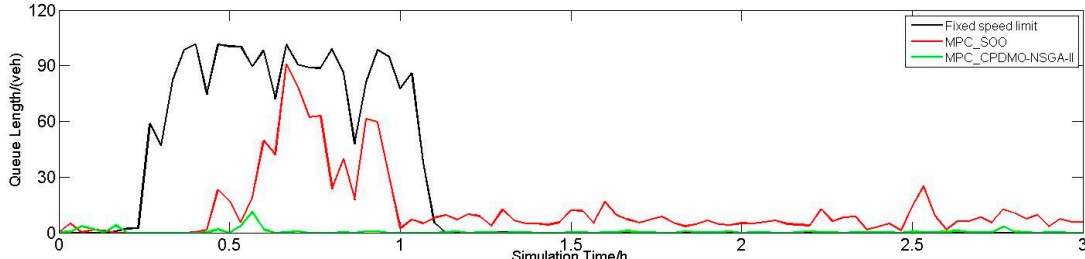

**Figure 13.** The queue length of the on-ramp with three control methods.

After applying MPC_SOO and MPC_CPDMO-NSGA-II, there is no queue of the mainline, so it is not plotted in Figure 13. Table 5 also proves the improvement, which lists average queue lengths of the on-ramp and its upstream mainline with three methods. In MPC_SOO, the queue of the on-ramp reduces by 45.8%. But the proposed method performs better, with which the average number of waiting vehicles declines to zero, indicating that almost all vehicles enter into the mainline.

**Table 5.** The queue length of the on-ramp and its upstream mainline with three control methods.

| Methods | Location | Mainline(veh) | On-ramp(veh) |
|---|---|---|---|
| **Fixed Speed Limit** | | 3 | 24 |
| **MPC_SOO** | | 0 (−100%) | 13 (−45.8%) |
| **MPC_CPDMO-NSGA-II** | | 0 (−100%) | 0 (−100%) |

## 7. Conclusions and Future Work

This paper mainly focuses on the freeway congestion control problem with the consideration of fuel consumption and emissions. It proposes a model predictive control method based on dynamic multi-objective optimization algorithm, MPC_CPDMO-NSGA-II to realize on-line traffic control. MPC_CPDMO-NSGA-II adopts dynamic multi-objective optimization to model predictive control to get more effective control solutions. Thus, this paper then explores an algorithm CPDMO-NSGA-II, which is an improvement of CPM_DMOEA. In order to augment convergence and reduce computation time, CPDMO-NSGA-II mainly revises original prediction models of centers and cluster shape. It also introduces and improves the method of determining historical centers on time series, which is based on reference points.

To verify the control method, this paper carries out a simulation research on a freeway with an on-ramp. The results show that it effectively improves efficiency of the freeway, and alleviates congestions, emissions and fuel consumption. Compared to MPC_SOO, the control solutions MPC_CPDMO-NSGA-II provided are more stable, so that it can ensure traffic safety to some extent. In addition, it can offer better solutions by balancing multiple objectives with conflicts. Thus, the vehicles of queue of the on-ramp and upstream mainline are significantly reduced. Although there are more vehicles the on-ramp released using MPC_CPDMO-NSGA-II, it does not cause congestions in the mainline. The efficiency of second segment even increases with VSL strategy. However, its efficiency of computation requires further study.

**Author Contributions:** Methodology, Project Administration, Supervision, Funding Acquisition, Writing—Original Draft Preparation, J.C.; Investigation, Methodology, Software, Validation, Writing—Review and Editing, Y.Y.; Data Curation, Investigation, Software, Validation, Visualization, Writing—Review and Editing, Q.G.

**Funding:** This research was funded by National Natural Science Foundation of China, grant number 61104166.

**Acknowledgments:** The authors would like to thank the National Natural Science Foundation of China (61104166).

**Conflicts of Interest:** We declare there are no conflict of interest regarding the publication of this paper. We have no financial and personal relationships with other people or organizations that could inappropriately influence our work.

## Appendix A. Acronym List

| Acronym List | |
| --- | --- |
| **English Abbreviations** | **English Full Name** |
| MPC | model predictive control |
| NSGA-II | a fast and elitist multi-objective genetic algorithm |
| CPDMO-NSGA-II | dynamic multi-objective optimization algorithm based on clustering and prediction with NSGA-II |
| MPC_CPDMO-NSGA-II | model predictive control based on CPDMO-NSGA-II |
| TOPSIS | Technique for Order Preference by Similarity to an Ideal Solution |
| VSL | variable speed limit |
| RM | ramp metering |
| AR, VAR | Autoregressive, Vector (multivariate) Autoregressive |
| RND, VAR, PRE, V&P | Random, Variation, Prediction, Variation and Prediction |
| CPM_DMOEA | clustering prediction model based dynamic multi-objective evolutionary algorithm |
| TTS | Total Time Spend |
| TTT | Total travel time |
| TWT | Total waiting time |
| TTD | Total Travel Distance |
| TE | Total Emissions |
| TF | Total Fuel Consumption |
| MPC_SOO | model predictive control based on single-objective optimization algorithm |

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
