# Peer review of "Freeway Traffic Congestion Reduction and Environment Regulation via Model Predictive Control"

_algorithms, doi:10.3390/a12100220_

Round 1
Reviewer 1 Report
See attachment

Author Response
However, the on-line optimization problem in model predictive control (MPC) are normally non-linear, non-convex and with complicated constraints. Well that is not totally true... these are actually very special cases since MPC is quite often design for linear or linarized system with linear constraints in order to get an easy to solve optimization problem (a QP basically).Answer: The authors feel sorry for not expressing the ideas clearly. Please see Line 95 to 98.
MPC is quite often designed for linear or linearized system with linear constraints in order to solve optimization problem easily. However, the on-line optimization problem in MPC with the characteristics of non-linearity, non-convex and complicated constraints is considered in some existed research recently.
Basic MPC literature is missing.
Answer: The authors feel sorry for missing basic MPC literature. Please see Line 85 to 86.
Model predictive control (MPC) is a new control algorithm proposed in the late 1970s. It is a heuristic control algorithm based on model prediction [3].
Do the authors refer to receding horizon when they talk about rolling horizon? If so, please use the correct terminology (receding horizon). If not, please explain what rolling horizon means.
Answer: The authors have referred to receding horizon when talk about rolling horizon, we have corrected this mistake. Please see Line 64, 453, 540, 543 and Figure 3.
After the literature review, the discussion jumps to the proposed algorithm. But the problem the authors intend to solve is not stated, so it is hard for the reader to understand what is the real proposal and why.
Answer: The authors have modified the overall structure of the paper by placing the model description part of the paper after literature review, followed by algorithm description, then problem description and simulation experiment. And delete section which aims to test the effectiveness of the proposed CPDMO-NSGA-II algorithm on the prospect of dynamic multi-objective optimization function, not the freeway control problem discussed in this paper. Please see Line 78 to 83.
The remainder of this paper is organized as follows. Section 2 provides the literature review. The macroscopic traffic flow model METANET, the VT-micro model and the integrated VT-macro model are described in Section 3. Section 4 discuss the proposed model predictive control method based on dynamic multi-objective optimization algorithm, MPC_CPDMO-NSGA-II. Section 5 describes the problem formulation in detail. Section 6 discuss the simulation result of the freeway control problem. Section 7 provides conclusions and future works.
It is very hard to follow Section 4, which is the main result of the work. The author should rewrite it with the intent of increasing readability and making the capture of the ideas easier for the audience.
Answer: The authors have rewritten Section 4 and adjusted the position of the algorithm in section 4.2.1 and put the main algorithm (CPDMO-NSGA-II) before the sub-algorithm (Cluster the individuals in the population and the Associate the individuals in the population), which makes the whole algorithm process more clearly.
English is very poor and so it is difficult to follow authors’ ideas.
Answer: The authors feel sorry for that so we have polished the overall English writing and grammar of the paper.
Reviewer 2 Report
Dear Authors
The manuscript proposes a novel predictive algorithm to improve general traffic aspects in a freeway. The subject is very interesting since transport is one of the most polluting and energy consuming activity. The results are promising and the manuscript is well structured.
There are still some points that the reviewer would like to clarify or correct in the manuscript before its publication:
Add an acronym list, please. The number of acronyms is very high and makes it necessary. Line 40: Reference METANET and VT-micro programs, please. Line 102: first word could be incorrect. The manuscript use references to other paper model or control notation: In order to make the manuscript more understandable add those acronyms to the acronyms table (Line 116: AR, VAR. Line 117: RND, VAR…, NSGA-II… Since the proposed algorithm is based on NSGA-II, a reference to the main NSGA-II paper should be added (Line 104?) Deb ; A. Pratap ; S. Agarwal ; T. Meyarivan “A fast and elitist multiobjective genetic algorithm: NSGA-II” IEEE Transactions on Evolutionary Computation ( Volume: 6 , Issue: 2 , Apr 2002 ) pp182-197. DOI: 10.1109/4235.996017 Line 132. Traditional MPC is typically multi-objective: minimum reference deviation, minimum variation or cost of some parameters vs minimum control action. In this case, the dynamic propriety of the DMOAs_MPC should be highlighted and the particularities of the studied system to consider it as a single-objective system should be explained. Line 137. What ‘environmental changes’ stand for in this context? Clarify or add some examples, please Lines 178, 192… Use the appropriate font size in the equation and variables in the manuscript Line 381, y ∈ {CO, HC, NOx, FC}. Why CO2 is not considered? Although proposed emissions have environmental effect, the most significant traffic emission, CO2, is not considered. Authors must discuss why CO2 has not been considered among the emissions. Line 396, table 4. Define τ, κ, υ… Use an appropriate font size. Consider that reference [3] is not a direct web search engine result. Line 472 ff. The manuscript proposes a comparison of the results of MPC_CPDMO-NSGA-II algorithm with two algorithms: first MPC_SOO and second one multiple objective weight sum method (line 477). The manuscript does not show the results of this weight sum method and only provide results for MPC_SOO and MPC_CPDMO-NSGA-II. Correct this point, (preferably) adding the results of the third method or removing it from the manuscript. Line 481: Which is the fixed speed limit? Line 491: ‘Figure 13(a)(b)(c) gives…’ Figure 13(d) is not mentioned although the text describes it. When discussing the results of the Figure 16, authors should provide a small discussion about the results in segment 3, even if they are not relevant enough to be added in the figure. Line 335. The document font and size have been changed in this line. Lines 481, 512, 531, 545. Consider the use of 5.4.1, 5.4.2… notation for the new sections. The control results do not provide discrete VSL values and present a potential variation time of 10s. This is not a classic situation in a freeway. Add a small discussion about this point either in the introduction of the manuscript or in the conclusions. Provide a small description about the RM control strategy.
Round 2
Reviewer 1 Report
The revised version of the manuscript is clearly an improvement w.r.t. the original one.
Comment:
I don't understand while the authors say that MPC is an heuristic control strategy. It is not at all!!!
Also, I miss more MPC literature, such as Rawlings's book, Camacho's book, Grune's book, etc
Author Response
I don't understand while the authors say that MPC is an heuristic control strategy. It is not at all!!!
Also, I miss more MPC literature, such as Rawlings's book, Camacho's book, Grune's book, etc
Answer: The authors feel sorry for making such a mistake, and we have deleted the word "heuristic". Please see Line 85 to 86.
In this paper, one of the highlights about MPC method are that a dynamic multi-objective genetic algorithm, named as CPDMO-NSGA-II is proposed to realize on-line optimization in MPC, and the proposed dynamic multi-objective genetic algorithm is a heuristic optimization algorithm.

Reviewer 2 Report
Dear Authors.
The reviewer wants to thank the authors the effort made to improve the manuscript, which at this moment should be more understandable for readers. There are still some small suggestions that the authors could consider before the manuscript publication:
Emphasize (in a positive way) the relation between CO2 emissions and the fuel consumption. For future studies try to include the CO2 emissions parameter in the model. 1st revision point 12: When using fixed speed limit strategy, have the authors define a speed limit in the model? Which is it? Degradation of the average speed regarding to speed limit could also indicate the benefits of the proposed model Please, review the sentence at line 493 and following. Reviewer finds it a bit confusing. 1st revision point 17. Authors have made an effort developing the control model for the freeway traffic. Although the model implementation could not be realistic nowadays: changing speed limit every 10 seconds to a not “round” number; it could be adequate under some situations like an autonomous car “environment”. The reviewer suggests to the authors to consider emphaticizing the possibilities of their study, in order to provide a more practical point of view. Line 466: “is1500m.”. A space is needed
Author Response
Emphasize (in a positive way) the relation between CO2 emissions and the fuel consumption. For future studies try to include the CO2 emissions in the model.Answer: The authors feel sorry for missing the explanation and have given it from Line 218 to 226.
VT-micro emission model does not mention the estimated value of emission rate. However, paper [1] shows that there is almost an affine relationship between fuel consumption and emission. And then emission can be computed as following:
(19)
Where and denote the emission rate and fuel consumption rate of vehicle for time step respectively, and are model parameters. Although is related to fuel consumption [23], the model considered in this paper is unfit for considering the environmental effect on the prospect of . It may take a lot of changes by considering it in the model, therefore, we should discuss it in the future study.
1st revision point 12: When using fixed speed limit strategy, have the authors define a speed limit in the model. Which is it? Degradation of the average speed regarding to speed limit could also indicate the benefits of the proposed model
Answer: The authors feel sorry for missing the explanation and have given the explanation. from Line 532 to 534.
In the whole simulation process, the fixed speed limit is directly implemented in VISSIM simulation environment, and minimum and maximum speed of the vehicle on each segment on the road is set as .
Please, review the sentence at line 493 and following. Reviewer finds it a bit confusing.
Answer: The authors feel sorry for missing the explanation and have given the explanation. Please see Line 506 to 514.
For comparison, a model predictive control based on single-objective optimization algorithm named as MPC_SOO, is used to solve the freeway control problem in this paper, and the single-objective optimization algorithm is genetic algorithm [28]. The performance indicator used in MPC_SOO is described as following:
(38)
Where are the weights, which are set as 0.4, 0.3, 0.3 respectively. To compare the result of the MPC_SOO algorithm and the MPC_CPDMO-NSGA-II algorithm under similar expert’s experience, the weights of the used in the MPC_SOO algorithm is set as the same value used in the TOPSIS method in the MPC_CPDMO-NSGA-II algorithm.
1st revision point 17. Authors have made an effort developing the control model for the freeway traffic. Although the model implementation could not be realistic nowadays: changing speed limit every 10 seconds to a not “round” number, it could be adequate under an autonomous car “environment”. The reviewer suggests to the authors to consider emphaticizing the possibilities of their study, in order to provide a more practical point of view.
Answer: The authors feel sorry for missing the explanation and have given it from Line 493 to 498.
Table 2 gives the value of parameter use in the MPC_CPDMO-NSGA-II algorithm. It can be seen from table 2 that the sampling period is set as 10s, the control period is set as 1min, the prediction horizon and the control horizon are set as 15min and 10min respectively in this paper. Since the control period is set as 1min, it means that the MPC_CPDMO-NSGA-II algorithm proposed in this paper is suitable for both the simulation experiment and on-line control on the small road network. For larger networks, since more computation time are required, other methods should be applied.
Line 466: “is1500m.” a space is needed
Answer: The authors have corrected this mistake. Please see Line 472 to 473.
The length of main road is 1500m.
